# Cross-sectional analysis of use of real-world data in single technology appraisals of oncological medicine by the National Institute for Health and Care Excellence in 2011–2021

Jiyeon Kang  ,[1,2] John Cairns  [1,2]

¹Department of Health Services Research and Policy, Faculty of Public Health and Policy, London School of Hygiene and Tropical Medicin, London, UK
²Centre for Cancer Biomarkers (CCBIO), University of Bergen, Bergen, Norway

**Correspondence to**
Dr Jiyeon Kang;
jiyeon.kang@lshtm.ac.uk

## ABSTRACT

**Objectives** This study aims to identify how real-world data (RWD) have been used in single technology appraisals (STAs) of cancer drugs by the National Institute for Health and Care Excellence (NICE).

**Design** Cross-sectional study of NICE technology appraisals of cancer drugs for which guidance was issued between January 2011 and December 2021 (n=229). The appraisals were reviewed following a published protocol to extract the data about the use of RWD. The use of RWD was analysed by reviewing the specific ways in which RWD were used and by identifying different patterns of use.

**Primary outcome measure** The number of appraisals where RWD are used in the economic modelling.

**Results** Most appraisals used RWD in their economic models. The parametric use of RWD was commonly made in the economic models (76% of the included appraisals), whereas non-parametric use was less common (41%). Despite widespread use of RWD, there was no dominant pattern of use. Three sources of RWD (registries, administrative data, chart reviews) were found across the three important parts of the economic model (choice of comparators, overall survival and volume of treatment).

**Conclusions** NICE has had a long-standing interest in the use of RWD in STAs. A systematic review of oncology appraisals suggests that RWD have been widely used in diverse parts of the economic models. Between 2011 and 2021, parametric use was more commonly found in economic models than non-parametric use. Nonetheless, there was no clear pattern in the way these data were used. As each appraisal involves a different decision problem and the ability of RWD to provide the information required for the economic modelling varies, appraisals will continue to differ with respect to their use of RWD.

## INTRODUCTION

Real-world data (RWD) are increasingly being used in the life cycle of drug development.[1] Clinical trials mainly generate the evidence to support decision-making in regulatory and health technology assessment (HTA), whereas RWD have been used complementarily

**STRENGTHS AND LIMITATIONS OF THIS STUDY**

⇒ This study investigated how real-world data (RWD) have been used in economic models of oncology medicines appraised by the National Institute for Health and Care Excellence based on 11 years of evidence.
⇒ This study systematically applied a specially developed data extraction protocol.
⇒ While the findings are likely to be less relevant to other disease areas, some insights with respect to the opportunities to use RWD and its limitations are transferrable.

for postmarketing surveillance. Interest in adopting and implementing RWD in the context of HTA has exploded in recent years[2] as the evidence gaps when relying heavily on clinical trials[3 4] are more widely recognised. Many HTA bodies view RWD as acceptable sources of data in the context of HTA.[5 6] In most European HTA organisations, RWD are accepted to inform epidemiological data, cost and resource use[7] and are used to fill evidence gaps by supporting the assumptions in economic models.[8] Health economics and outcome research organisations, such as the European Network for HTA and the International Society for the Professional Society for Health Economics and Outcomes Research, have been active in proposing how RWD can be collected and analysed in the HTA context.[9–12]

The National Institute for Health and Care Excellence (NICE) has also shown increasing interest in RWD.[13] NICE has highlighted the value of RWD and wants to use RWD to improve understanding of healthcare, resolve gaps in knowledge and drive early access to innovations.[14] In June 2022, NICE introduced a real-world evidence (RWE) framework for the more extensive use of RWD in

producing NICE guidance.[15] NICE anticipates that RWD can reduce some of the uncertainties arising from limitations of randomised controlled trial (RCT) evidence, such as uncertain generalisability, the need to make indirect comparisons and limited long-term follow-up. There is a common ground that RWD can provide useful information in challenging circumstances, such as with small patient populations, rare diseases and cases where robust evidence is lacking, including single-arm trials.[5 16]

Although the use of RWD in HTA appears to be relatively new, the use of RWD is not entirely new in NICE. RWD have already been used in diverse ways in NICE appraisals when developing their guidance. For example, NICE issued technology appraisal (TA) guidance for rituximab for follicular non-Hodgkin's lymphoma (TA226). In this TA, Government Actuary's Department life tables, based on the mortality experience of a population, were used for age and gender-adjusted mortality. More recently, registry or hospital data, commonly regarded as RWD, have been used for economic modelling. TA guidance for lenalidomide with rituximab for previously treated follicular lymphoma (NICE TA627) used UK registry data for the clinical outcomes of the comparators due to the absence of data from direct treatment comparison.[17] Another example is European Chart Review data used to make indirect treatment comparisons when comparative data were absent in an appraisal of ibrutinib for treating Waldenstrom's macroglobulinaemia (NICE TA795).[18]

The use of RWD in NICE appraisals has been reviewed in several studies. A review of NICE guidance published in 2015 and 2016 found limited use but increasing prominence of RWD.[19] Also, RWD have been used to predict long-term effectiveness in the submissions to different HTA agencies.[20] Several studies have discussed the role of RWD in supporting healthcare decision-making and have broadly outlined the benefits and challenges of using RWD.[21–23] However, these studies considered how RWD have been used in HTA decision-making based on case studies. One study reviewed the use of RWD in economic models in single technology appraisals (STAs) of cancer drugs more comprehensively.[24] It found that RWD were extensively used in the cost-effectiveness analyses. However, the ability of this study to answer the question 'How have RWD been used?' was limited since it only reported the number of times RWD were used as model inputs in appraisals.

To date, no study has shown how the use of RWD has changed over time or varied by type of cancer. Also, the extent to which RWD have been used to supplement the evidence in economic models has not been fully reviewed. This remains a gap in our understanding of the use of RWD. This study aims to provide a more complete picture of use of RWD in economic models for cancer drug appraisals. The specific purposes of this study are to identify trends in the use of RWD over time and by type of cancer and to characterise patterns of use of RWD in economic models.

## METHODS

Data were extracted from 229 NICE STAs of oncology medicines for which NICE issued guidance between January 2011 and December 2021, following a protocol developed to document information about using RWD in economic models in NICE STAs of oncological medicines.[25] Extracted data include general information about STAs, evidence-specific information such as characteristics of primary clinical evidence and the use of RWD in the economic model. Information on the use of RWD was extracted for both the base case and sensitivity analyses. However, only the use of RWD in the base case is highlighted in this paper. The definition of RWD in this study was data relating to patient health status and/or the delivery of healthcare routinely collected from non-experimental settings.[25] The protocol extensively covers the components of economic models where RWD can be potentially used. While reporting the number of uses of RWD in each appraisal is a simple way to describe the use of RWD, this approach has limitations in understanding trends in the use of RWD in NICE STAs. A fuller picture of how RWD have been used can be obtained by examining similarities and differences in use of RWD between appraisals. The pattern of use of RWD records which elements of the economic evaluation are informed by RWD. The data extraction protocol distinguishes 31 areas in economic modelling where RWD can be used (online supplemental file 1). These 31 areas are only considered as components for economic modelling in this paper. The use of RWD in these different areas in each appraisal was identified and reviewed for common patterns.

Different uses of RWD and patterns are characterised by which components of an economic model were informed by RWD. There are three categories of use: any, non-parametric and parametric use of RWD. Any use of RWD refers to its use in any part of the economic model regardless of how RWD were used. Parametric use means that RWD provide numerical values for specific variables in the economic model. For example, using data to estimate overall survival (OS) or resource use in the economic model is categorised as parametric use. Non-parametric use is where RWD are used to develop the economic model structure and to support assumptions in the model. Examples of non-parametric use include using RWD to select comparators or to validate the choice of survival distribution. Thus, the label non-parametric implies that the RWD do not provide a parameter for the economic model, and is not referring to non-parametric methods. This categorisation groups the uses and increases focus on where RWD were used in the economic model, and facilitates comparison of the use of RWD across appraisals, for example, patterns of use over time and by type of cancer. Also, this categorisation enables examination of the association between the number of non-parametric use and the number of parametric use. When data are identified and used in synthesising evidence, the data can be used in multiple ways. Spearman's rank-order correlation was

carried out to test whether these two different ways of using RWD were associated.

Major and minor uses of RWD were identified based on which part of the economic evaluation they informed. Among 31 components included in the extraction protocol, three components (OS of intervention/comparators, volume of treatment of intervention/comparators, choice of comparators) were identified as major uses of RWD, which are highly likely to influence the estimated incremental cost-effectiveness ratio. This assumption was supported by experts' opinions.[26] The remaining components were considered minor uses of RWD. This study highlights the sources of RWD used in these three major components to provide detailed information about the potential and limitations of using various sources of RWD. Common sources of RWD identified in the NICE RWE framework were used for this study: electronic health records, administrative data, registries, chart reviews and observational cohorts with primary data collection. Where the sources were not defined in the framework, they were classified in separate categories: namely data from the Office for National Statistics (ONS) and market share data. While the framework classifies health surveys, interviews and focus group as RWD, this study excluded them, following the definition of RWD, if the data are not routinely collected.

### Patient and public involvement
None of the patients or the public were involved in any part of this research.

### RESULTS
#### The use of RWD in economic models
Figure 1 shows how use of RWD has changed 2011–2021. RWD were used in the economic models of most STAs. Parametric use of RWD was greater than non-parametric use. Over the 11 years, an average of 76% of appraisals made parametric use of RWD, whereas an average 41% of appraisals used RWD non-parametrically.

Figure 2 reports use of RWD by type of cancer. There was some difference between the non-parametric use and parametric use in the different types of cancer. In some appraisals, there was no non-parametric use of RWD, whereas for some types of cancer all appraisals made parametric use of RWD. These appraisals were found in the cancers with only a few STAs (thyroid cancer, genomic biomarker-based cancer, pancreatic cancer and neuroblastoma). Among cancer types where there were more than three STAs, the appraisals of skin cancer showed the most marked difference between the two uses. All the appraisals of skin cancer made parametric use of RWD. In comparison, non-parametric uses of RWD were made in only 27.3% of these appraisals.

The relationship between the two different uses was reviewed. Figure 3 shows the distribution of parametric use and non-parametric use of RWD per appraisal. In 43% of appraisals, parametric use was made when no non-parametric use was made. With respect to the association, no clear relationship is apparent in the heatmap. Spearman's rank correlation shows that non-parametric use and parametric use are not statistically associated ($r_s$=0.07, p=0.28).

#### Patterns of use of RWD
Identified patterns of use of RWD, regardless of the type of use, are presented in table 1. Patterns of use of RWD observed on a single occasion are not separately identified in the table but are grouped under *Others*. No dominant pattern of use of RWD in economic models was identified in these appraisals. Among identified patterns of using RWD (n=117), only 15 patterns appeared in more than two appraisals, cumulatively 51% of all appraisals. 16%

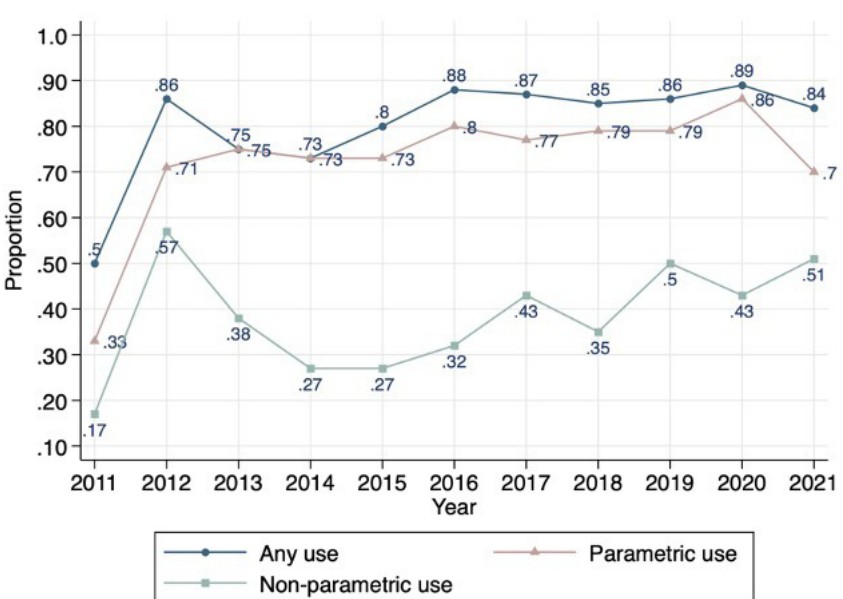

**Figure 1** Parametric use and non-parametric use of real-world data (RWD) over time.

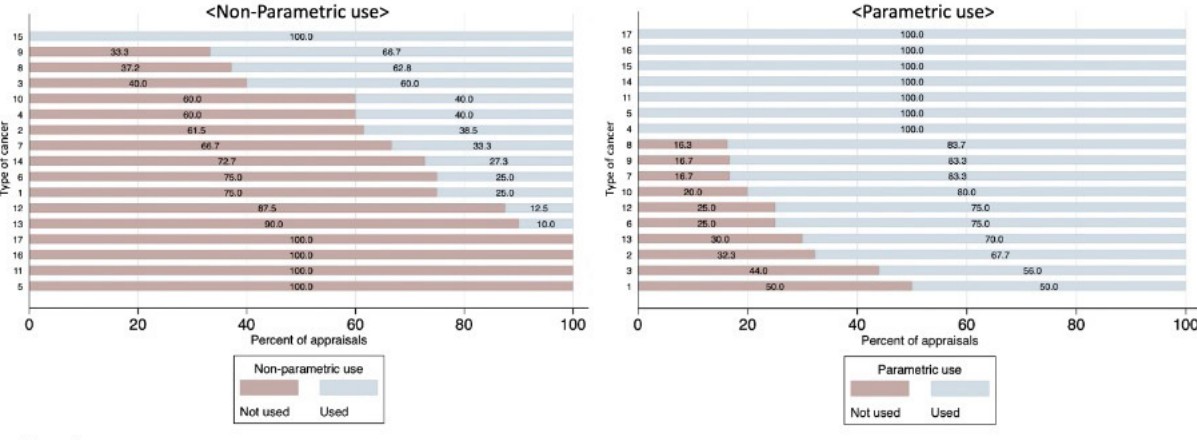

**Figure 2** Non-parametric use and parametric use of real-world data (RWD) by type of cancer.

of STAs did not use RWD in any part of the economic model. The second most commonly observed pattern was the pattern estimating OS of intervention and comparators (6% of all patterns). The third most common pattern was use of RWD to estimate end-of-life resource use (5% of all patterns).

Non-parametric use and parametric use of RWD were separately reviewed (online supplemental file 2). Three-fifths of all appraisals made no non-parametric use of RWD. The most common pattern of non-parametric use of RWD was to validate the choice of survival distribution for the intervention and comparators (9% of appraisals), followed by use of RWD to identify comparators (6% of appraisals). The patterns of parametric use of RWD were more diverse than the patterns of non-parametric use. 76% of appraisals used RWD to inform at least one

parameter in the economic model. Using RWD for estimating end-of-life resource use was the most common pattern (10% of appraisals), followed by the use of RWD to estimate OS for the intervention and comparators (7% of appraisals).

### Sources of the RWD

Figure 4 summarises the sources of RWD used for the major components of the economic evaluation. As sources of data are likely to be different for intervention and comparators in estimating OS and volume of treatment, the use of RWD was reviewed separately for the intervention and the comparators. Depending on the components, frequently used sources of RWD were different. Data from ONS were commonly used for estimating OS, whereas market share data were frequently

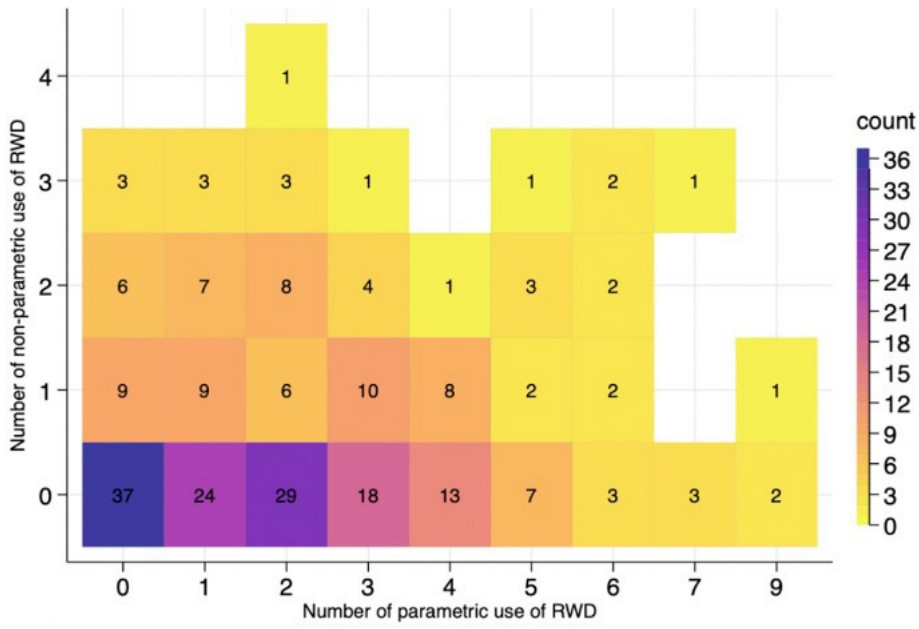

**Figure 3** Parametric use and non-parametric use of real-world data (RWD) by appraisal.

**Table 1** Distribution of patterns of use of real-world data

| Patterns | n (%) |
|---|---|
| No use of RWD | 37 (16.16) |
| Estimating overall survival of intervention and comparators | 13 (5.68) |
| Estimating resource use of end of life | 12 (5.24) |
| Estimating resource use of end of life and health state cost | 8 (3.49) |
| Estimating resource use of health state cost | 7 (3.06) |
| Estimating overall survival of intervention and comparators and resource use of end of life and health state cost | 6 (2.62) |
| Estimating overall survival and progression-free survival of intervention and comparators and resource use of health state cost | 5 (2.18) |
| Validating survival distribution of intervention and comparators and estimating resource use of end of life | 5 (2.18) |
| Estimating overall survival and progression-free survival of intervention and comparators | 5 (2.18) |
| Estimating resource use of end of life and dose adjustment of intervention and comparators | 4 (1.75) |
| Estimating the volume of treatment of intervention and comparators | 3 (1.31) |
| Estimating overall survival of intervention and comparators and resource use of health state cost | 3 (1.31) |
| Validating survival distribution of intervention and comparators | 3 (1.31) |
| Choosing comparators | 3 (1.31) |
| Choosing comparators and estimating resource use of health state cost | 3 (1.31) |
| Others | 112 (48.9) |
| Total | 229 (100) |

RWD, real-world data.

a source of RWD for determining relevant comparators in economic models. These data were often collected by the company or private international health information entities such as IQVIA. Three sources of RWD (registries, administrative data, chart reviews) were found across the three major components. On average, 22% of sources of RWD used for major components were registry data. The Surveillance, Epidemiology, and End Results Program and Flatiron were frequently used registry data. Administrative data were the source for 15% of the occasions that RWD were used for major components.

Clinician surveys or expert opinions are often used in economic models when determining the resource use or distribution of subsequent treatments. For example, the appraisal of nivolumab for previously treated unresectable advanced or recurrent oesophageal cancer (NICE TA707) used a clinician survey to determine the frequency of resource use in order to estimate disease management costs.[27] However, data from clinician surveys or expert opinions are not classified as RWD following the definition used in this study.

## DISCUSSION

This study identified how RWD have been used in economic models in NICE appraisals of oncology medicines over time and by type of cancer, focusing on the specific ways that RWD have been used and patterns of RWD use. Most previous studies investigated the use of RWD in small samples of appraisals without explicit criteria for identifying RWD or tried to identify its use from stakeholder interviews. This study extracted explicit and systematic information on the use of RWD and assessed the use of RWD in economic models in NICE STAs of oncology medicines.

When looking at the use of RWD over time, it is notable that RWD have been used at least since 2011. Regardless of data types, it is standard practice to use the best available evidence in NICE TAs.[28] The key aim of NICE appraisals is to provide a summary of the relevant evidence to ensure that the appraisal committee can make fully informed decisions about the new technology.[29] If RCTs offer limited information for decision-making, the evaluation may be strengthened by using alternative data, such as observational data. For this, it is important to use such data correctly. There is a NICE decision support unit (DSU) report and a technical support document to assist better analysis and interpretation of treatment effects from non-RCT studies.[16 30] These documents guide the analysis of non-RCT data and observational data more generally to avoid potential biases, such as selection bias in the estimation of treatment effects. This highlights NICE's long-standing interest in using diverse sources of evidence for drug appraisals.

Having documented where RWD have been used, the uses were classified by what was informed by RWD. This study found that although RWD were widely used across the economic models they were less often used non-parametrically. This result can be reviewed with the findings about the sources of RWD in the three major components. Among identified diverse sources of RWD, registries and administrative data were used in a minority of appraisals. It might be challenging to identify evidence matching the population specified in the decision problem. Clinical experts' opinions were often used to justify assumptions regarding the non-parametric variables. Expert judgement, including expert opinions or expert elicitation, can be helpful when relevant data are unlikely to exist, such as for rare cancers. However, inconsistent application and insufficient reporting of the process are concerns.[31 32]

Along with expert opinions, RWD available in the National Health Service could provide more systematic information to describe routine practice better.

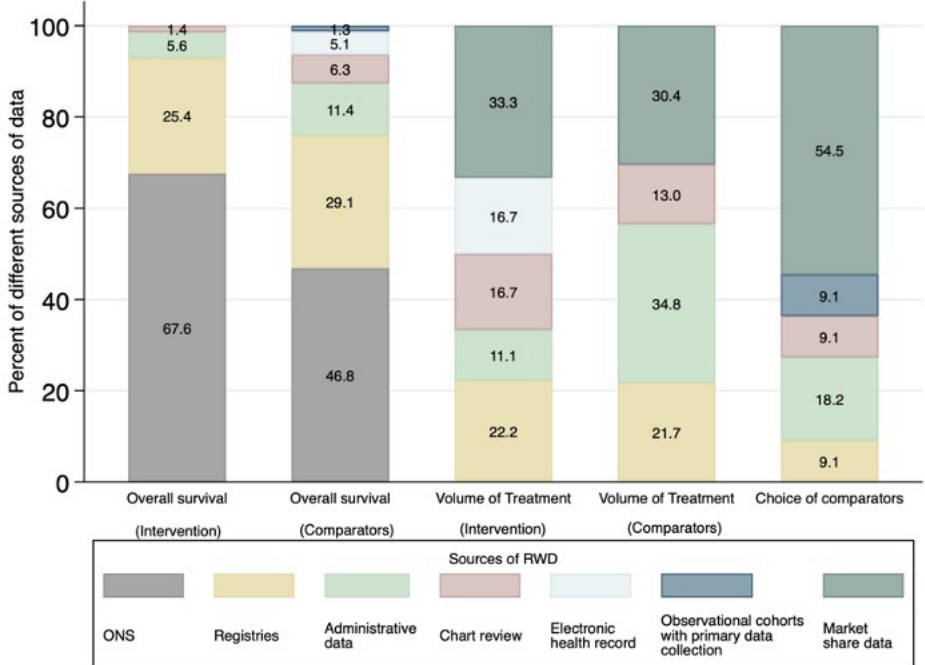

**Figure 4** Sources of real-world data (RWD) used in three major components of the economic evaluation. ONS, Office for National Statistics.

An example is the Cancer Drugs Fund review appraisal of pembrolizumab (NICE TA766).[33] In the original appraisal, the subsequent treatment was one of the sources of uncertainty. The data on subsequent treatments from clinical trials were incomplete and did not reflect UK clinical practice. In the review appraisal, systemic anticancer therapy dataset was used to identify subsequent treatment after adjuvant pembrolizumab treatment, and clinicians supported the findings of the data. The appraisal committee concluded that the data, despite its immaturity, reflected practice. Although this cannot be generalised to all cases, for example, immature data with expert's opinions might reduce uncertainty regarding long-term survival, this example shows that in some cases, RWD and expert opinion can provide more robust evidence to support decision-making.

The correlation between the use of RWD for parametric and non-parametric purposes was investigated. Interestingly, no significant correlation was found. It was expected that manufacturers might be willing to use RWD for non-parametric purposes when some parameters are estimated based on RWD. Similarly, fewer non-parametric uses are expected if manufacturers do not need to use RWD as a source of parameter values. However, the statistical results and the heatmap suggested no significant association between uses. This result may imply that the different sources of RWD differ in their ability to address uncertainties in the economic model. RWD used for non-parametric purposes might not contain enough information to inform parameters in the economic model. For example, market share data can help identify comparators by showing which treatment is frequently used, whereas such data cannot indicate the volume of treatment due

to the missing information about treatment sequences. Moreover, the evidence in the manufacturer's submission is often identified through a literature review. The identified literature can provide some information required for the model in the manufacturer's submission. However, it may not be sufficient to provide all the information required in different part of economic models since literature only reports its outcome to answer its own research questions. If manufacturers more often directly explore RWD databases when developing their models, the association between parametric use and non-parametric use will increase.

In this study, the most common pattern was no use of RWD and there was no dominant pattern among patterns involving at least one use of RWD. This suggests that there is no agreed approach to using RWD in STAs. Each appraisal raises a different mix of decision-making issues. Also, the ability of RWD to address a particular decision problem varies by appraisal context. Consequently, approaches and opportunities to incorporate supplementary data such as RWD will vary across appraisals. Although there was no dominant pattern of non-parametric use, about half of the identified patterns included uses of RWD to validate or corroborate the survival distributions for either the intervention or the comparators. This may be a direct consequence of the NICE DSU document issued in 2011 (last updated in 2013) that proposed that the model used to extrapolate survival could be justified using external data sources.[34] Using RWD to validate or corroborate the survival distributions can be viewed as an attempt to validate the clinical plausibility of survival models using external data.

This study has focused on the use of RWD in the base case modelling of cost-effectiveness. Data on the use of RWD in sensitivity analyses and the results are presented elsewhere.[26] The use of RWD as supplementary data in sensitivity analyses was expected to be more common than in the base case since the remaining uncertainty around parameters in an economic model is usually explored in sensitivity analyses using alternative evidence.[35] However, parametric use of RWD in the sensitivity analysis was found in only a few appraisals. The interviews, conducted for the separate study,[26] help draw an implication. The manufacturer is less likely to present the RWD analysis results in the sensitivity analysis if there is no clear evidence that additional RWD are favourable to its outcome. Processing RWD can require significant resources in terms of collection and analysis. Suppose there is no incentive to use RWD in a sensitivity analysis. In that case, manufacturers prefer to use other published RCTs to explore the uncertainty in their model inputs and the survival distribution. There are a few cases where companies submit additional RWD in response to technical engagement or when forced to with a negative appraisal consultation document. However, this brings other problems as the quality of the data can be very poor, and reporting is not transparent.

While RWD could provide information reflecting routine clinical practice, the reliability of the RWD can be questioned in some cases. RWD are expected to provide information reflecting routine practice. However, depending on the timing and context of data collection, RWD may not accurately reflect routine practice. Due to changes in clinical practice or the small number of patients recruited, the RWD sample is potentially different from the whole population. In TA660, darolutamide with androgen deprivation therapy for treating hormone-relapsed non-metastatic prostate cancer,[36] RWD were used to estimate healthcare resource by the target population. The external evidence review group was concerned that the study population was recruited over a wide time interval, which may have seen substantial changes in clinical practice, and that the sample was too small to accurately identify the clinical benefits since the primary outcome was obtained from 44 patients diagnosed with the specific indication. This example highlights that RWD do not always reflect current practice. How and what information was collected to answer the question is critical to the use of RWD in STAs.

This study has some limitations. The main one is that the information about the use of RWD was extracted from the manufacturer's submission. The extracted data did not necessarily reflect appraisal committee's preferences regarding sources of evidence. The data in the manufacturers' submissions are primary evidence in the STA process. However, appraisal committees do not necessarily accept the data presented by manufacturers. Their preferences regarding RWD could be different from those of the manufacturer. A study reviewing the appraisal committee's preferences regarding RWD is in progress and may provide a more comprehensive understanding of the previous use of RWD in NICE STAs. Despite this limitation, this study contributes to understanding the use of RWD in appraisals. It is the first study to provide systematic information about use of RWD in economic modelling by reviewing appraisals of oncology drugs over 11 years.

RWD data can and have been used in the economic models, but to date there is no consistent or systematic way of using RWD. While the use of RWD in designing and validating economic models and helping to reduce uncertainty has been emphasised,[15] this study found that non-parametric uses were less common than parametric uses and there were no dominant patterns of using such data. The NICE RWE framework can play a key role, such as encouraging more systematic use of RWD and identifying when RWD can reduce uncertainties in economic model for drug appraisals. Despite the potential benefits of this framework, the extent to which it will facilitate the use of RWD to reduce uncertainty is unclear. A recent study found that RWD collected while cancer drugs were provided through a managed access scheme in England had done little to reduce uncertainty.[37] Further in-depth study on use of RWD in NICE appraisals could inform a discussion of the opportunities and limitations of the NICE RWE framework.

## CONCLUSION

NICE has had a long-standing interest in the use of RWD in STAs. A systematic review of oncology appraisals suggests that RWD have been widely used in diverse parts of the economic model. Between 2011 and 2021, parametric use was more commonly found in economic models than non-parametric use. Nonetheless, there was no clear pattern in the way these data were used. As each appraisal involves a different decision problem and the ability of RWD to provide the information required for the economic modelling varies, appraisals will continue to differ with respect to their use of RWD.

**Contributors** Both authors contributed to conceptualising and designing the study. JK (the guarantor) contributed to acquisition of the data and data analyses. JK and JC contributed to interpretation of the data. JK wrote the original version of the manuscript. JC contributed to the critical revision of the manuscript. JK has full responsibility for the work and conduct of the study.

**Funding** JK is supported by the Centre for Cancer Biomarkers, University of Bergen, funded by the Research Council of Norway (grant number: 223250).

**Disclaimer** The funder is not involved in any aspect of the study conduct or the decision to submit the paper for publication.

**Competing interests** None declared.

**Patient and public involvement** Patients and/or the public were not involved in the design, or conduct, or reporting, or dissemination plans of this research.

**Patient consent for publication** Not applicable.

**Ethics approval** This study was approved by the Ethics Committee of the London School of Hygiene and Tropical Medicine on 14 November 2019 (17315).

**Provenance and peer review** Not commissioned; externally peer reviewed.

**Data availability statement** Data are available in a public, open access repository. All data used for this research are publicly available and can be accessed through

the National Institute for Health and Care Excellence website [https://www.nice.org.uk/guidance/published?ngt=Technologyappraisalguidance&ndt=Guidance].

**Open access** This is an open access article distributed in accordance with the Creative Commons Attribution 4.0 Unported (CC BY 4.0) license, which permits others to copy, redistribute, remix, transform and build upon this work for any purpose, provided the original work is properly cited, a link to the licence is given, and indication of whether changes were made. See: https://creativecommons.org/licenses/by/4.0/.

**Author note** The original protocol for the study. The protocol for this study is published: Kang J, Cairns J. Protocol for data extraction: how real-world data have been used in the National Institute for Health and Care Excellence appraisals of cancer therapy. *BMJ Open.* 2022;12:e055985. doi: 10.1136/bmjopen-2021-055985.

**ORCID iDs**
Jiyeon Kang http://orcid.org/0000-0002-1587-9674
John Cairns http://orcid.org/0000-0001-6442-0440

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
