## [Reviewer comments · BMJ Open]

ARTICLE DETAILS

TITLE (PROVISIONAL)	A cross-sectional analysis of use of real-world data in single technology appraisals of oncologic medicine by the National Institute for Health and Care Excellence in 2011 – 2021
AUTHORS	Kang, Jiyeon; Cairns, John

VERSION 1 – REVIEW

REVIEWER	van Asselt, Antoinette University Medical Center Groningen, Epidemiology
REVIEW RETURNED	12-Nov-2023

GENERAL COMMENTS	This is a thorough and elaborate work, well written, of interest to a selected audience It would have been really interesting to see not only how and how frequent RWD is used in appraisals but how valid and acceptable the RWD is evaluated to be, and typically what critique the ERG and committee have. I understand this probably is the topic of another (next) paper, but because this paper is only descriptive, its value may be limited because of that. Comments The terminology can be confusing. The authors talk of patterns of use, types of use, elements, components (in text and in tables). What is exactly patterns of use? Definition of parametric and non-parametric? The exact meaning of these terms and what they contain mostly becomes clear after the methods section. It would greatly improve readability and ease of interpretation if the definitions are made very clear at the start and then maintained consistently throughout the paper. Relationship between parametric and non-parametric use – the hypothesis the researchers made for the potential associations here only become clear in results/discussion. In the methods section though where the analysis is explained, the sense and logic of this analysis is not very obvious to this reader. And in all honesty, it is not that relevant altogether maybe whether there are any associations or not. P13 of 32, lines 6-25 – the example of pembrolizumab. RWD in this case was used to describe subsequent treatments, and the committee thought the RWD to reflect practice despite its immaturity. The message that may come across from describing it like this, is that using immature RWD can be a useful tool in appraisals. I would say that very heavily depends on how the RWD is used. For describing subsequent treatments yes, it may be quite useful. For extrapolating survival data, maybe not so useful. I think it is very important to emphasize that here.
---

	P5 of 32 lines 36-40 rituximab with lenalidomide (2020) may not be the best example to demonstrate that the use of RWD in NICE is not entirely new, because it is a fairly recent STA. P14 of 32 lines 54 slightly surprising to only mention the sensitivity analyses bit in the discussion section. Even though it is the subject of another paper, it would be good to announce a bit earlier on in the paper P16 of 32 line 16 'the committee' not all readers will immediately understand what committee exactly this is about. Please make explicit.
--	---

VERSION 1 – AUTHOR RESPONSE

Reviewer: 1	Response
This is a thorough and elaborate work, well written, of interest to a selected audience. It would have been really interesting to see not only how and how frequent RWD is used in appraisals but how valid and acceptable the RWD is evaluated to be, and typically what critique the ERG and committee have. I understand this probably is the topic of another (next) paper, but because this paper is only descriptive, its value may be limited because of that.	Thank you for your comments on this paper. Following your comments, we revised the manuscript. With respect to the critique by the ERG and committee, we agree that it would be more interesting to include their views. However, the main objective of this paper is to describe the current use of RWD in appraisals of cancer drugs in more detail. We try to keep this manuscript readable by focusing on the main objective. Analysis of critiques of ERG and committee is another topic for the following paper. In the discussion, we addressed it as a limitation of this paper and briefly mentioned future study.
The terminology can be confusing. The authors talk of patterns of use, types of use, elements, components (in text and in tables). What is exactly patterns of use? Definition of parametric and non-parametric? The exact meaning of these terms and what they contain mostly becomes clear after the methods section. It would greatly improve readability and ease of interpretation if the definitions are made very clear at the start and then maintained consistently throughout the paper.	We added texts to provide further information about patterns of use and components (p.6 L10-17). We also understand that 'parametric' and 'non-parametric' might be confusing as these terms are widely used in statistics. We made a note to clarify that these terms have different meanings from their uses in statistics (P7 L12-14).
Relationship between parametric and non-parametric use – the hypothesis the researchers made for the potential associations here only become clear in results/discussion. In the methods section though where the analysis is	We added some words to clarify the purpose of the Spearman rank-order correlation:

explained, the sense and logic of this analysis is not very obvious to this reader. And in all honesty, it is not that relevant altogether maybe whether there are any associations or not.	“Also, this categorisation enables examination of the association between the number of non-parametric and the number of parametric uses. When data are identified and used in synthesising evidence, the data can be used in multiple ways. Spearman rank-order correlation was carried out to test whether these two different ways of using RWD were associated.” (p.7 L11-15)
P13 of 32, lines 6-25 – the example of pembrolizumab. RWD in this case was used to describe subsequent treatments, and the committee thought the RWD to reflect practice despite its immaturity. The message that may come across from describing it like this, is that using immature RWD can be a useful tool in appraisals. I would say that very heavily depends on how the RWD is used. For describing subsequent treatments yes, it may be quite useful. For extrapolating survival data, maybe not so useful. I think it is very important to emphasise that here.	We revised the paragraph: “Although this cannot be generalised to all cases, for example, immature data with expert’s opinions might reduce uncertainty regarding long-term survival, this example shows that in some cases, RWD and expert opinion can provide more robust evidence to support decision-making.” (p.13 L1-4)
P5 of 32 lines 36-40 rituximab with lenalidomide (2020) may not be the best example to demonstrate that the use of RWD in NICE is not entirely new, because it is a fairly recent STA.	We added an example of rituximab for follicular non-Hodgkins lymphoma (TA226). “For example, NICE issued technology appraisal (TA) guidance for rituximab for follicular non-Hodgkins lymphoma (TA226). In this TA, Government Actuary’s Department life tables, based on the mortality experience of a population, were used for age and gender-adjusted mortality.” (p.4 L13-16) This TA is a relatively early appraisal, and RWD were used in the economic modelling.
P14 of 32 lines 54 slightly surprising to only mention the sensitivity analyses bit in the discussion section. Even though it is the subject of another paper, it would be good to announce a bit earlier on in the paper	We added the text: “Information on the use of RWD was extracted for both the base-case and sensitivity analyses. However, only the use of RWD in the base-case is highlighted in this paper.” (p.6 L5-6)

P16 of 32 line 16 'the committee' not all readers will immediately understand what committee exactly this is about. Please make explicit.	We changed the wording to "appraisal committee."
---	--

VERSION 2 – REVIEW

REVIEWER	van Asselt, Antoinette University Medical Center Groningen, Epidemiology
REVIEW RETURNED	15-Feb-2024

GENERAL COMMENTS	Authors have responded in an adequate fashion to the questions raised. As said before, the fact that the manuscript does not contain information ERG's and committee's views and critiques of the RWD probably limits the interest of the paper to part of the audience.
--